# Dehydration entropy drives liquid-liquid phase separation by molecular crowding

Sohee Park [1], Ryan Barnes[2], Yanxian Lin[3], Byoung-jin Jeon[4], Saeed Najafi[2,5], Kris T. Delaney[5], Glenn H. Fredrickson[4,5,6], Joan-Emma Shea[2,7], Dong Soo Hwang[1,8✉] & Songi Han [2,6✉]

Complex coacervation driven liquid-liquid phase separation (LLPS) of biopolymers has been attracting attention as a novel phase in living cells. Studies of LLPS in this context are typically of proteins harboring chemical and structural complexity, leaving unclear which properties are fundamental to complex coacervation versus protein-specific. This study focuses on the role of polyethylene glycol (PEG)—a widely used molecular crowder—in LLPS. Significantly, entropy-driven LLPS is recapitulated with charged polymers lacking hydrophobicity and sequence complexity, and its propensity dramatically enhanced by PEG. Experimental and field-theoretic simulation results are consistent with PEG driving LLPS by dehydration of polymers, and show that PEG exerts its effect without partitioning into the dense coacervate phase. It is then up to biology to impose additional variations of functional significance to the LLPS of biological systems.

[1] Division of Environmental Science and Engineering, Pohang University of Science and Technology (POSTECH), 77 Chengam-ro, Nam-gu, Pohang 37673, Republic of Korea. [2] Department of Chemistry and Biochemistry, University of California, Santa Barbara, CA 93106, USA. [3] Department of Biomolecular Science and Engineering, University of California, Santa Barbara, CA 93106, USA. [4] Materials Department, University of California, Santa Barbara, CA 93106, USA. [5] Materials Research Laboratory, University of California, Santa Barbara, CA 93106, USA. [6] Department of Chemical Engineering, University of California, Santa Barbara, CA 93106, USA. [7] Department of Physics, University of California, Santa Barbara, CA 93106, USA. [8] Division of Integrative Biosciences and Biotechnology, Pohang University of Science and Technology (POSTECH), 77 Chengam-ro, Nam-gu, Pohang 37673, Republic of Korea.
✉email: dshwang@postech.ac.kr; songi@chem.ucsb.edu

Around 50% of the protein sequence with segment length >30 amino acids coded by the human genome are predicted to be intrinsically disordered proteins (IDPs) without a three-dimensional structure[1,2]. Recent research has provided clues that IDPs play a key role in protein regulation inside cells[3–5], as well as participate in the formation of membraneless organelles[6–8]. Interestingly, some membraneless organelles composed of IDPs have displayed liquid-like physical properties[9–12], suggesting that intracellular droplet formation by liquid–liquid phase separation (LLPS) may be a relevant mechanism for the formation of membraneless organelles. There are examples in the literature of membraneless organelles found in living cells[10,13,14].

Historically, complex coacervation (CC), which results in LLPS, has been suggested to share the genesis of the protocell[15,16]. This is because coacervate can be composed of simple components, while they are capable of taking up various substances and segregate from the environment. Given that CC represents one of the most robust mechanisms to drive LLPS, it has been used as a model system to investigate whether their formation involving IDPs correlate with human disease conditions[17–21]. If LLPS with IDPs is to be a regulatory state of importance to cellular processes, it makes sense that its formation and dissolution conditions be modulated by physiological relevant factors within crowded environments[22].

Complex coacervation is a phenomenon in which polyelectrolytes separate into a polyelectrolyte-rich phase (dense phase) and a polyelectrolyte-depleted phase (dilute phase)[23]. CC typically occurs when oppositely charged polyelectrolytes (referring to either the entire biopolymer or a biopolymer segment) interact with each other by electrostatic attraction, and ultimately form polyelectrolyte microdroplets termed the complex coacervate phase. Coacervation can also occur by an inter-molecular association of a single component, known as simple coacervation[24–26]. Interactions other than electrostatic interactions have also been shown to modulate or even drive LLPS, including by cation-π[27] or hydrophobic interaction[28,29]. Complex coacervation is affected by many factors including ionic strength, pH, polyelectrolyte concentration, a balanced mixing ratio of oppositely charged polyelectrolytes, molecular weight of the polyelectrolytes, as well as temperature and the crowding pressure[30]. It is by now firmly established that CC, fundamentally and without specific biological driving factors, is an equilibrium state that can be described by a phase diagram[22,31–33]. Thus, the above-listed factors all contribute to modulating the free energy for CC formation ($\Delta G_{CC}$), where CC will occur when $\Delta G_{CC}$ ($=\Delta H_{CC} - T\Delta S_{CC}$) is negative. For many CC processes $\Delta H_{CC}$ is a small value, but the entropy gain may be positive ($\Delta S_{CC} > 0$), in which case lower critical solution temperature (LCST) behavior is observed, where increasing temperature favors LLPS[18,22,34]. The origin of the entropy gain is assumed in the literature to be due to counterion-release[35–38]. However, a recent study relying on experimental and field-theoretic simulations (FTS) of CC between the IDP tau and RNA demonstrated that counterion-release to be a negligible driver of CC, or at least does not need to be invoked to replicate LCST-driven CC[22], while hydration water-release may be a major contributor to entropy gain. In a study by Van der Gucht, counterion-release entropy was found to be most negative at the lowest ionic strength[37]. However, it is important to note that both counterion-release and dehydration entropy would be greater at lower ionic strength where the effective surface charge of the polyelectrolytes is greater.

The question we ask is what are the driving forces for CC between oppositely charged polyelectrolytes? What is the minimum requirement to establish LCST (or UCST) behavior in

CC? The CC between polyelectrolytes with minimal sequence complexity and hydrophobicity compared to protein will teach us about the base property of CC, especially under conditions that mimic the cellular environment. A cell constitutes a high concentration of biomacromolecules and so its internal environment is crowded (80−400 mg mL$^{-1}$)[39–41]. The CC stability under intracellular conditions requires stability of electrostatically driven CC under physiological ionic strength[18,42], unless other factors are at play. We have empirical evidence that molecular crowding is a key factor that stabilizes the CC of IDPs under cellular conditions. The question is whether this is due to a base property of CC or rather due to some specific properties of the involved IDPs, and what is the underlying mechanism of crowding-stabilized CC. Crowding reduces the effective volume available to the biomolecular constituents, and thus affect molecular interactions and reactions[43]. Polyethylene glycol (PEG) is often used to mimic the intracellular crowding environment in vitro, as it reduces the effective volume for other biomolecular constituents[44–47], by attracting water and so dehydrating the other constituents[48,49]. However, how PEG promotes CC mechanistically, including whether PEG mainly acts to increase the excluded volume or interacts with and partitions in the dense coacervate phase, is not known.

In this paper, we choose ε-poly-L-lysine (εPL) and hyaluronic acid (HA) as the model constituents for CC, to specifically focus on the questions whether: (1) PEG increases the yield of coacervate, (2) PEG partitions into the coacervate phase or not, (3) εPL-HA CC displays entropy-driven LCST behavior, and (4) PEG induces dehydration of the polyelectrolyte constituents and/or of the coacervate phase.

We use a wide range of experimental tools and an advanced computer simulation method termed FTS[50–52] to understand the influence of PEG on εPL-HA LLPS. Experimentally, we characterize key properties of the coacervate phase, including the dynamics of interstitial water in the coacervate phase by pulsed-field gradient (PFG) and of surface water hydrating the polyelectrolyte constituents by overhauser dynamic nuclear polarization (ODNP). We examine the CC stability as a function of temperature. We measure the viscosity by microrheology and the interfacial tension of the coacervate droplet by image analysis of droplet coalescence, as well as the molecular dynamics of the polyelectrolytes by fluorescence recovery after photobleaching (FRAP) within the coacervate phase, in the absence vs. presence of PEG. Computationally, we rely on a numerical FTS tool that fully accounts for fluctuations and can compute structure and thermodynamic properties without approximations[53–55], and hence allow us to compute the density distribution of coarse-grained polyelectrolytes and PEG in the dense and dilute phase upon LLPS-CC.

## Results

**General properties of εPL-HA CC with/without PEG.** Positively charged εPL (Supplementary Fig. 1a) and negatively charged HA (Supplementary Figure 1b) in 0.1 M sodium acetate buffer (pH 5.0) were mixed, and NaCl added at a series of concentration of 0, 10, 20, 30, 40, 50, 60, and 200 mM to generate the εPL-HA coacervate. Mixing of εPL and HA initially resulted in the formation of CC microdroplets, and the microdroplets were condensed to a macroscopic coacervate phase by centrifugation (Fig. 1a). To quantify the coacervate phase, the turbidity of the precursor microdroplet coacervate suspension and volume fraction ($V_{dense}/V_{total}$) of the macrophase-separated coacervate were evaluated (details are described in "Methods"). To observe the effects of PEG on various properties of CC, 10% (w/v) PEG was added to the sample.

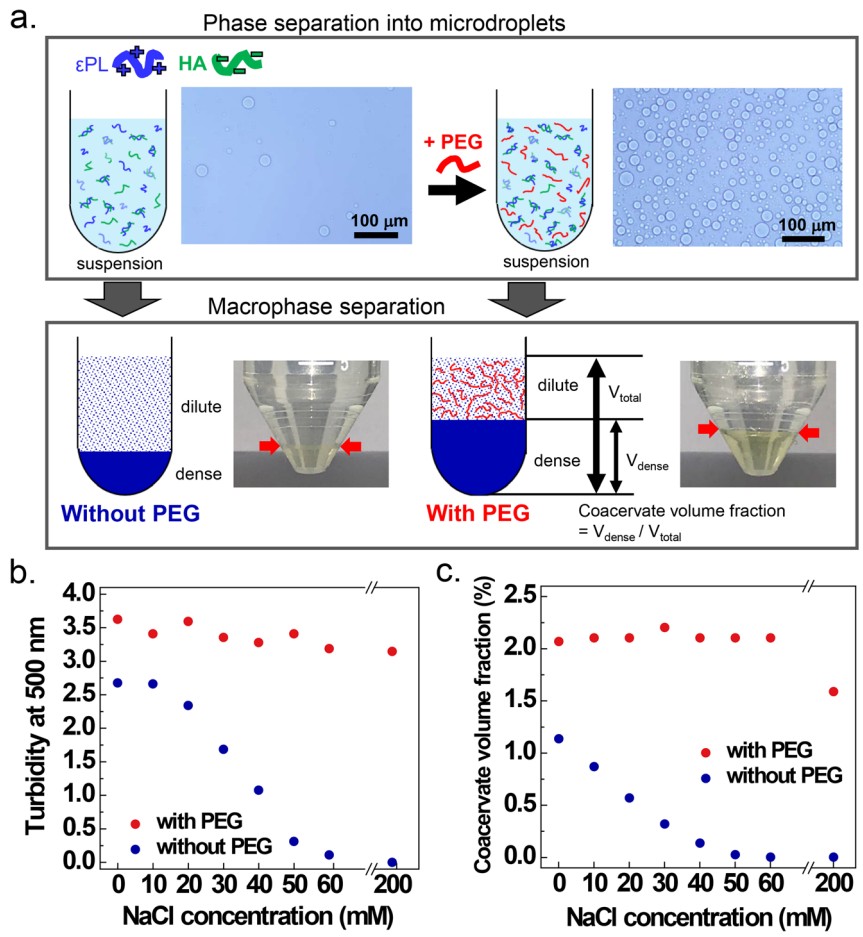

**Fig. 1 Schematic and general properties of εPL-HA complex coacervation. a** Schematic of εPL-HA complex coacervation (0.1 M, pH 5.0 sodium acetate, additional 40 mM NaCl) with and without PEG. The microdroplet coacervate suspension was observed by an optical microscope, and the macrophase separation occurred after centrifugation. **b** Turbidity (at 500 nm) of the suspension and (**c**) coacervate volume fraction as a function of additional NaCl in 0.1 M sodium acetate (pH 5.0).

The first question is whether PEG alters the yield of the coacervate phase. A microdroplet coacervate suspension with vs. without 10% (w/v) PEG in the presence of 40 mM NaCl was separately observed under the light microscope. More microdroplets were observed by eye in the presence of PEG (Fig. 1a). Furthermore, relative turbidity and volumetric analysis of the macrophase-separated coacervate phase were performed. The turbidity and the coacervate volume fraction decreased with increasing NaCl concentration in the absence of PEG; this result is expected for complex coacervating systems[33,56,57]. In contrast, the coacervate phase formed in the presence of PEG was found to be invariantly stable with increasing salt concentration, even in the presence of 200 mM NaCl (Fig. 1b, c). This observation confirms that PEG increased the coacervate yield and stability.

Next, is simply a higher coacervate quantity obtained upon addition of PEG, or is the density of the dense coacervate phase also altered? We observed that the density of the dense CC phase increased in the presence of PEG ($1.16 \, \mathrm{g \, mL^{-1}}$) compared to without PEG ($1.00 \, \mathrm{g \, mL^{-1}}$) (Table 1). We conclude that PEG increased not only the coacervate yield and stability, but also the coacervate density.

How does PEG increase the coacervate density? Is it because PEG partitions into the dense coacervate phase, or is it because more polyelectrolytes (εPL and HA) get packed into the coacervate phase? To answer this question, the polyelectrolyte concentration in the macro-separated dense and dilute phase was estimated (details are described in "Methods"). The polyelectrolyte concentration of the dense phase formed without PEG of $188 \, \mathrm{mg \, mL^{-1}}$ was ~20 times higher than the dilute phase formed without PEG of $7.9 \, \mathrm{mg \, mL^{-1}}$, whereas the concentration of the dense phase formed with PEG of $321 \, \mathrm{mg \, mL^{-1}}$ was ~100 times higher than that of the dilute phase formed with PEG of $3.3 \, \mathrm{mg \, mL^{-1}}$ (Table 1). In other words, PEG increased the polyelectrolytes density in the dense coacervate phase by an additional twofold by extracting the polyelectrolytes constituents from the dilute phase, while the total polyelectrolyte mass can be accounted for in the CC phase upon addition of PEG.

Does PEG partition into the dense phase? We performed [1]H NMR of the macro-separated dilute and dense phases in the presence and absence of PEG. The peak at 3.7 ppm[58], signifying protons of the PEG repeating unit ($-O-CH_2-CH_2-$), was detected only in the dilute phase formed with PEG (Fig. 2). These results unambiguously show that PEG does not directly partition in the dense phase, but is exclusively solubilized in the dilute phase. In another view, εPL-HA CC with PEG may promote a mixture of associative and segregative phase separation[59]. Associative phase separation, e.g. CC, is driven by attraction between the biopolymers, while segregative phase separation, e.g. gelatin/dextran[60], is promoted by an effective repulsion between the biopolymers. Based on these concepts, εPL-HA CC can be interpreted as being promoted by associative phase separation

**Table 1 Concentration of the polyelectrolytes in the dilute and dense phases.**

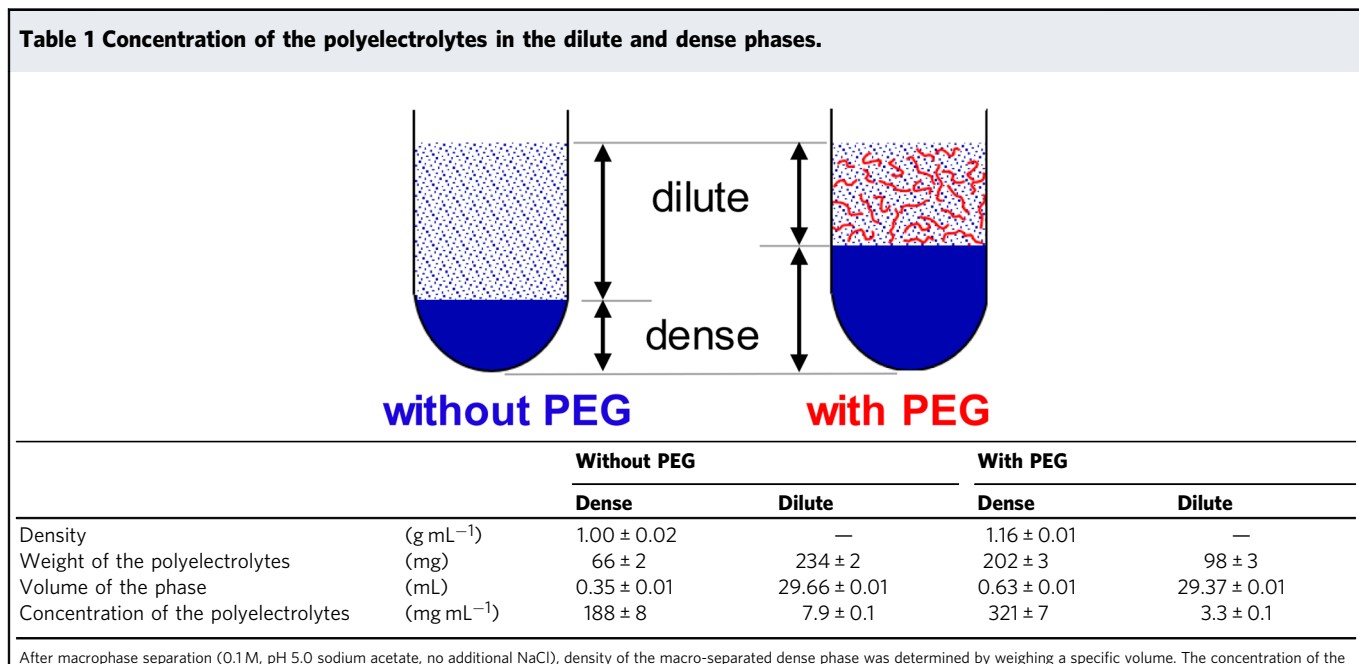

| | | Without PEG | | With PEG | |
|---|---|---|---|---|---|
| | | **Dense** | **Dilute** | **Dense** | **Dilute** |
| Density | (g mL$^{-1}$) | 1.00 ± 0.02 | — | 1.16 ± 0.01 | — |
| Weight of the polyelectrolytes | (mg) | 66 ± 2 | 234 ± 2 | 202 ± 3 | 98 ± 3 |
| Volume of the phase | (mL) | 0.35 ± 0.01 | 29.66 ± 0.01 | 0.63 ± 0.01 | 29.37 ± 0.01 |
| Concentration of the polyelectrolytes | (mg mL$^{-1}$) | 188 ± 8 | 7.9 ± 0.1 | 321 ± 7 | 3.3 ± 0.1 |

After macrophase separation (0.1 M, pH 5.0 sodium acetate, no additional NaCl), density of the macro-separated dense phase was determined by weighing a specific volume. The concentration of the polyelectrolyte in each phase was calculated by the weight of the polyelectrolytes in the phase and volume of the phase. All values expressed as (mean ± standard deviation).

driven by the attraction of εPL and HA, and the exclusion of PEG due to the segregative phase separation between εPL-HA complexes and PEG. We can conclude that PEG significantly increased the yield and density of the dense phase, without directly partitioning into the dense phase (Fig. 3), suggesting that the role of PEG is that of dehydrating the polyelectrolyte constituents, and providing an additional entropic benefit for CC formation.

**Entropy-driven εPL-HA CC with/without PEG.** Is εPL-HA CC entropy driven? A signature of an entropy-driven process is that elevated temperature facilitates the process. This would be reflected in the CC displaying LCST. To check the formation of CC at different temperatures, a mixed solution of εPL and HA (with additional 60 mM NaCl) in the presence and absence of PEG was placed on a temperature-controlled stage of an optical microscope, and studied at different temperatures. In the absence of PEG, no coacervate was found at 25 °C under the microscope, but small microdroplets were observed at 50 °C, while the number of droplets increased as the temperature increased to 75 °C (Fig. 4a). To further substantiate the effect of temperature on facilitating CC, the absorbance of the suspension was measured with increasing temperature (Fig. 5). The graph shows that εPL-HA CC follows LCST behavior (Fig. 5a, black line), corroborating the hypothesis that entropy gain enhances εPL-HA CC.

What is the effect of PEG on the LCST trend of CC? In the presence of PEG (10% (w/v)), the microdroplets (formed under additional 60 mM NaCl) were observed even at 25 °C and the temperature rise (to 80 °C) further triggered the coalescence of the droplets which increased the size of the droplets (Fig. 4b). When 1% (w/v) PEG was added, the absorbance reached a maximum at a much lower temperature than in the absence of PEG, while upon addition of 10% (w/v) PEG the absorbance already reached a maximum value at RT (Fig. 5a). In other words, the addition of PEG in and of itself has the effect of increasing the entropy gain for CC, i.e. $\Delta S_{CC} > 0$ upon addition of PEG that drives the CC system towards an even more negative $\Delta G_{CC}$ at elevated temperatures.

Why is the addition of PEG and elevated temperature displaying the same effect of enhancing CC? Increased temperature tends to release hydration water by breaking water−water and/or water−polyelectrolyte hydrogen bonds, promoting CC by increasing $\Delta S_{CC}$. In fact, CC requires partial removal of surface-bound hydration water around the polyelectrolytes for the polyelectrolytes to interact with others, to share hydration water and form polyelectrolyte complexes. The same rationale may apply to PEG that induces the weakening of the water−water and/ or water−polyelectrolyte hydrogen bonds by attracting water to its own hydration shell. Hence, increasing temperature and PEG both weaken the water−water and water−polyelectrolytes hydrogen bonds of hydration water of the polyelectrolyte constituents, facilitate dehydration and so entropically promotes CC. At even higher temperature, even PEG will lose its solubility in water[61], but in the present temperature regime outcompetes the polyelectrolytes for water.

Consequently, maximum absorbance should be observed in the presence of a threshold amount of PEG as dehydration has already occurred before increasing the temperature. Curiously, the absorbance of the coacervate suspension with the addition of 10 and 20% (w/v) PEG seemed to decrease with high temperature. However, careful observations showed that this trend is not due to the disappearance of CC at high temperature. Rather, this apparent decrease in absorbance with increasing PEG beyond a threshold value is because macrophase separation occurred faster, and the supernatant was more transparent at 70 °C than at RT when the coacervate suspension with 10% (w/v) PEG was incubated for 2 h at RT or 70 °C (Fig. 5b). Also, the decrease in absorbance accelerated as temperature increased (Fig. 5c). Therefore, the decrease in absorbance of the suspension with PEG at high temperature was a result of fast droplet coalescence that accelerated the separation of the bulk phases. In addition, we measured the coacervate volume fraction at different temperatures (RT, 40, and 70 °C), and found the coacervate volume fraction to be insignificantly changed (Supplementary Table 2). In short, a threshold amount of PEG not only dehydrated the polyelectrolyte constituents to promote the phase separation to form microdroplets, but also accelerated the macrophase separation, confirming that the role of PEG is that of dehydrating the polyelectrolyte constituents.

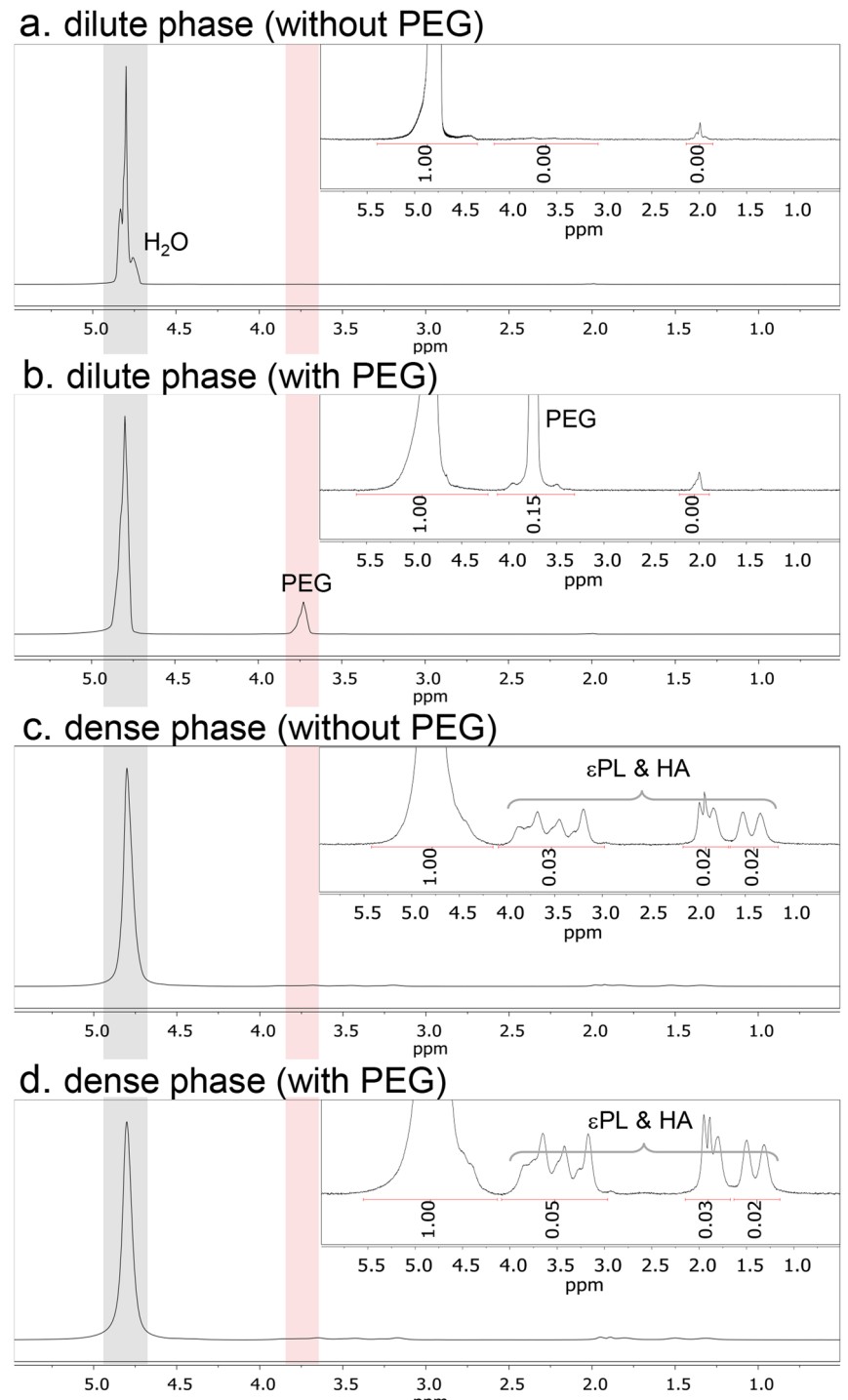

**Fig. 2 [1]H PFG NMR full and zoom-in (50-fold amplification) spectra.** The spectrum from macro-separated **a** dilute phase (without PEG), **b** dilute phase (with PEG), **c** dense phase (without PEG), and **d** dense phase (with PEG). Complex coacervation occurred in 0.1 M, pH 5.0 sodium acetate buffer, with no additional NaCl.

**Dehydration in εPL-HA CC with/without PEG.** We had previously suggested that dehydration might be the main contributor to entropy gain to cause CC, but what type of water is being released? To clarify this, we focus on the study of interstitial and hydration water around polyelectrolyte constituents. The diffusivity of interstitial water was determined by [1]H PFG-NMR, and that of hydration water by ODNP. PFG-NMR, also known as the pulse field gradient spin echo (PGSE) NMR, is a well-known technique to measure self-diffusion of small

molecules in solution[62–64]. In contrast, ODNP selectively amplifies [1]H NMR signal of adjacent water molecules by transfer of polarization from electron spins of the spin label to [1]H nuclear spin of water molecules through electron-[1]H dipolar coupling, where the efficiency of dipolar coupling-driven electron-[1]H cross relaxation is affected by the proximity and movement of adjacent water molecules. Therefore, ODNP has been a powerful approach to quantify hydration water translational diffusion dynamics near the spin label that is covalently

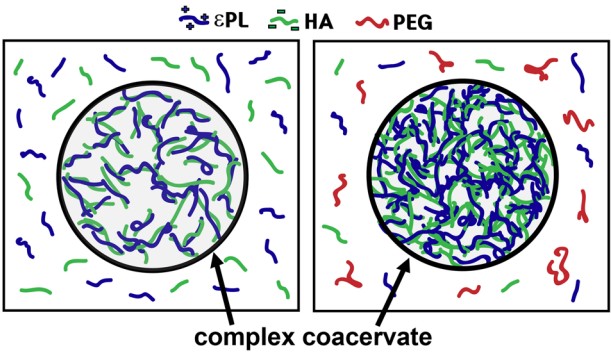

**Fig. 3 Illustration of the εPL-HA complex coacervation with and without PEG.** PEG did not directly participate in the complex coacervation but increased the coacervate yield and density.

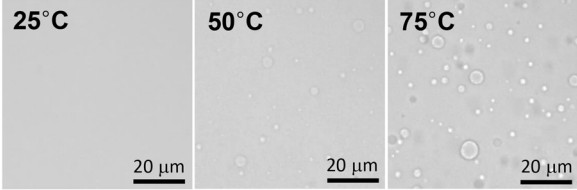

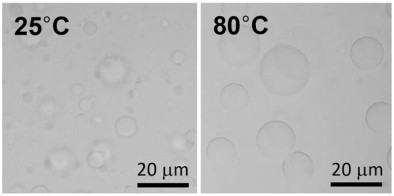

**Fig. 4 Microscope images of the εPL-HA microdroplet suspension.** **a** Without PEG and **b** with 10% (w/v) PEG at specific temperature (under 0.1 M, pH 5.0 sodium acetate, additional 60 mM NaCl).

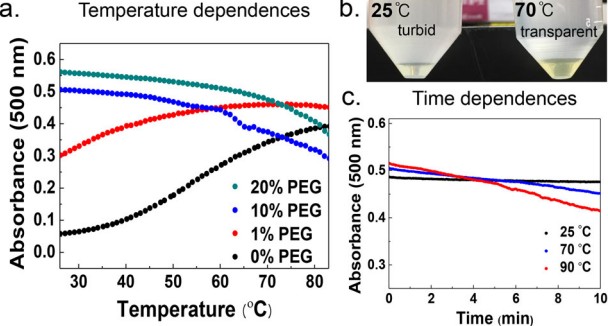

**Fig. 5 Temperature dependences of the εPL-HA complex coacervation.** **a** The absorbance of the εPL-HA microdroplet suspension (0.1 M, pH 5.0 sodium acetate, additional 60 mM NaCl) with different PEG% (w/v) in a function of temperature. **b** Photo of macrophase separation after 2 h incubation of the micro-separated coacervate suspension with 10% PEG at 25 and 70 °C. **c** Time effect on the εPL-HA complex coacervation in the presence of PEG at 25, 70 and 90 °C.

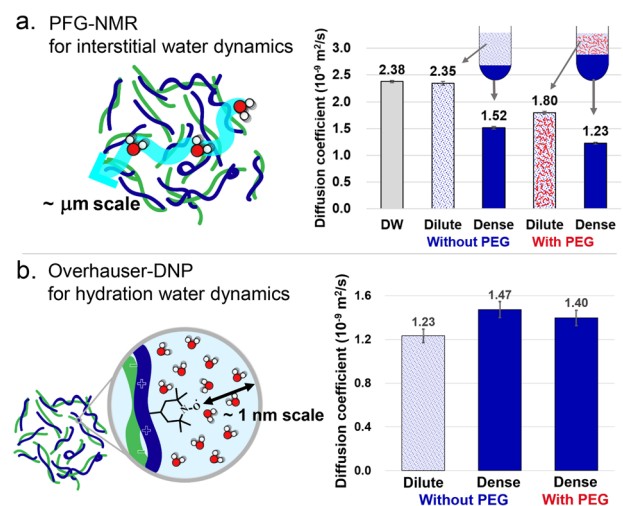

**Fig. 6 Water dynamics. a** Diffusion coefficients of interstitial water in the macro-separated phases (0.1 M, pH 5.0 sodium acetate, without additional NaCl) were measured by PFG-NMR. Error bars indicate the standard deviation. **b** Diffusion coefficients of hydration water were obtained by Overhauser-DNP. Error bars with the 5% deviation on the means have been marked to show the effect of experimental uncertainty.

bound to specific surface sites[65–67]. Both techniques can provide water diffusion coefficients, but there is a difference in length scale over which movement is detected. PFG-NMR detects water movement in the range of a micrometer to tens of micrometers, whereas ODNP detects water movement in the range of sub-nanometer around the spin label.

It would be natural to imagine that interstitial water in the dense phase would be slow due to high polyelectrolyte density. The diffusivity of interstitial water in the macro-separated dense and dilute phases in the absence of PEG was measured by PFG-NMR. The diffusivity was $2.35 \times 10^{-9}$ m$^2$ s$^{-1}$ in the dilute phase and $1.52 \times 10^{-9}$ m$^2$ s$^{-1}$ in the dense phase, in the absence of PEG (Fig. 6a). The diffusivity of interstitial water in the dilute phase was almost the same as that in deionized water (DW) ($2.38 \times 10^{-9}$ m$^2$ s$^{-1}$). The diffusivity was slightly decreased (by 60%) in the dense phase compared with the dilute phase, i.e. surprisingly of the same order of magnitude as in DW, although the concentration of the polyelectrolytes in the dense phase was 20 times higher than in the dilute phase. Therefore, we can say that interstitial water was still freely diffusing in the highly concentrated coacervate phase. The relatively fast movement of interstitial water in the dense phase indicates that water–water interaction and the water–coacervate complex interaction are relatively weak.

The dynamics of hydration water, closely interacting with the polyelectrolytes within 1 nm of its surface, was our next target.

First, we investigated the rotational motion of spin label by electron paramagnetic resonance (EPR) lineshape analysis of the macro-separated dense and dilute phases containing spin-labeled εPL. The rotational motion of the spin label was not changed in the macro-separated dense phase (Supplementary Fig. 3), implying that the polyelectrolytes in the dense phase were dynamic without notable restriction. Next, the diffusivity of hydration water in the macro-separated dilute and dense phases was determined by ODNP. The diffusivity of hydration water was $1.23 \times 10^{-9}$ m$^2$ s$^{-1}$ in the dilute phase and $1.47 \times 10^{-9}$ m$^2$ s$^{-1}$ in the dense phase in the absence of PEG (Fig. 6b). Interestingly, the diffusivity of hydration water was slightly increased in the dense phase, which may be due to partial dehydration of hydration water near the polymer surface that reduces diffusion retardation of water near the dehydrated polymer surface. Hence, this result is consistent with the concept that hydration water-release to bulk water contributes entropy gain for CC formation. In addition, the higher diffusivity of hydration water in the dense phase reflects on the weakened water–polyelectrolyte interaction.

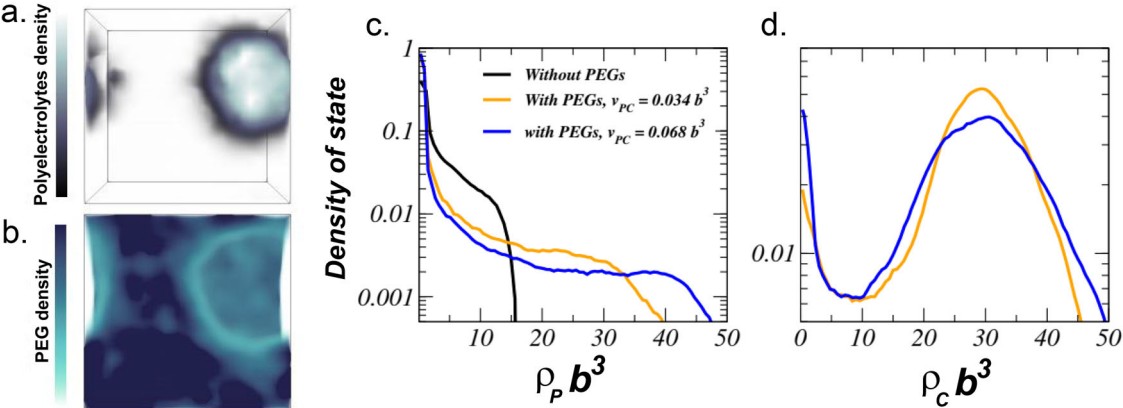

**Fig. 7 FTS results.** The density distribution of **a** polyelectrolytes and **b** PEG in a two-phase solution in which a dilute supernatant and dense coacervate phase of polyelectrolytes coexist. The density distribution (normalized histogram) of **c** polyelectrolytes and **d** PEG. In this system, we set the total density of polyelectrolytes and PEG at $\rho_P b^3 = 1.5$ and $\rho_C b^3 = 15$. The black, orange and blue curves show the density distribution of polyelectrolytes without PEG and with PEG at cross-excluded volume strengths $v_{PC} = 0.034b^3$ and $0.068b^3$, respectively.

Still, what is the role of PEG on the interstitial water of dense CC? According to the PFG-NMR results, interstitial water was slower in the presence of 10% (w/v) PEG than in the absence of PEG. The macro-separated dilute phase was slowed down from $2.35 \times 10^{-9}$ to $1.80 \times 10^{-9}\,\mathrm{m^2\,s^{-1}}$, and the macro-separated dense phase was slowed down from $1.52 \times 10^{-9}$ to $1.23 \times 10^{-9}\,\mathrm{m^2\,s^{-1}}$ (Fig. 6a). The diffusivity of interstitial water in the dilute phase decreased accordingly as the concentration of PEG increased (Supplementary Fig. 4), while we know that all PEG stayed in the dilute phase (Supplementary Fig. 5). Therefore, the slowing of interstitial water by PEG in the dilute phase is the direct effect of PEG acting as a viscogen. However, the slowing of interstitial water by PEG in the dense phase is noteworthy as PEG stayed outside of the dense CC phase. This might be an effect of increased polyelectrolyte density upon addition of PEG (Table 1). To directly investigate whether PEG changed the amount of water in the dense phase, we measured water content in the dense phase formed with and without PEG by a moisture analyzer. The water content was $81.18 \pm 0.01\%$ (w/w) in the dense phase formed without PEG, and $72.33 \pm 0.01\%$ (w/w) in the dense phase with PEG (Supplementary Table 1). This means that PEG lowered the water content of the dense phase by about 10%. We can conclude that PEG extracted interstitial water into the dilute phase (dehydration of interstitial water) upon CC.

The next question is what is the role of PEG on the surface hydration water of the polyelectrolytes upon CC? The diffusivity of hydration water was $1.40 \times 10^{-9}\,\mathrm{m^2\,s^{-1}}$ in the dense phase formed with PEG, i.e. a comparable value to that of hydration water in the dense phase formed without PEG ($1.47 \times 10^{-9}\,\mathrm{m^2\,s^{-1}}$) (Fig. 6b). This indicates that the property of hydration water around polyelectrolytes upon CC is not affected by PEG. Given that our temperature dependence (Fig. 5a) represents that adding PEG shows the same effect as the effect of temperature increase (causing partial dehydration of hydration water), we conclude that PEG extracts the interstitial, bulk-like, water from the CC phase without significantly altering the surface hydration properties of the polyelectrolytes. This is consistent with the nearly unaltered polyelectrolyte spin label dynamics and the weak interacting nature of the polyelectrolytes in the dense CC phase.

**Field-theoretic simulation study of CC.** The experimental observations are consistent with the hypothesis that PEG exerts its effect by dehydrating water from the dense into the dilute phase, while mainly partitioning in the dilute phase. However, modeling of this effect is critical to test the main mechanism

of action of PEG in promoting CC. We performed fully fluctuating FTS using complex Langevin sampling to elucidate the relative importance of PEG as a molecular crowder on CC using a coarse-grained bead-spring model of polyelectrolytes[68]. Our system has a volume $V$ and contains $n_1$ polycations and $n_2$ polyanions (we set $n_1 = n_2 = n$ to satisfy charge neutrality), with a degree of polymerization $N_P = 25$. The average polyelectrolyte density is thus $\rho_P = n_P N_P V^{-1}$, where $n_P = 2n$. PEG in our system is modeled as a noncharged polymer by employing the same bead-spring model. The degree of polymerization of PEG chains is $N_C = 100$ with density $\rho_C = n_C N_C V^{-1}$, where $n_C$ is the number of PEG chains in solution. Each bead on the polyelectrolytes carries a charge $\pm 1$ in units of the elementary charge $e$, and successive beads are connected by harmonic bonds with root-mean-square separation $b$. Additionally, the monomers interact via two nonbonded potentials in a representation of implicit water: a short-range repulsive excluded volume interaction and a long-range electrostatic interaction that is described by the Coulomb potential $u(r) = l_B r^{-1}$, where $l_B$ is the Bjerrum length defined by $l_B = e^2(4\pi\varepsilon_0\varepsilon k_B T)^{-1}$ and $\varepsilon$ is the dielectric constant of water. While the Bjerrum length can be modulated with both temperature and added salt, here we fix it at a typical value of $l_B = b$. With regard to the short-ranged potential, polyelectrolytes and PEG beads interact with their own species through a soft repulsive excluded volume parameter $v_{PP}$ and $v_{CC}$, respectively. For the sake of simplicity, we assumed that $v = v_{PP} = v_{CC}$, with $v = 0.0068b^3$, where $b$ is the statistical segment length. As a proxy for dehydration propensity, we modulate the cross-excluded volume interaction between the monomers of the polyelectrolytes and PEG ($v_{PC}$). Further details of FTS can be found in Supplementary Methods.

FTS conducted with the model just described indeed predict CC formation, as depicted in an instantaneous snapshot (Fig. 7a, b). It is observed that both polyelectrolyte density and PEG density vary considerably between the coexisting dilute and dense phases. The polyelectrolyte density distribution obtained by a thermally averaged histogram analysis of the density profiles within the simulation cell is presented (Fig. 7c). At the coexistence of dilute and dense domains, PEG was found to enhance the CC by driving the polyelectrolytes from the dilute phase into the dense region, and so increase the packing density. Furthermore, in the presence of PEG, the density of polyelectrolytes in the dilute and dense region was found to be dictated by the strength of the cross-excluded volume interaction that is a control factor over the polyelectrolyte dehydration propensity. The increase of cross-excluded volume

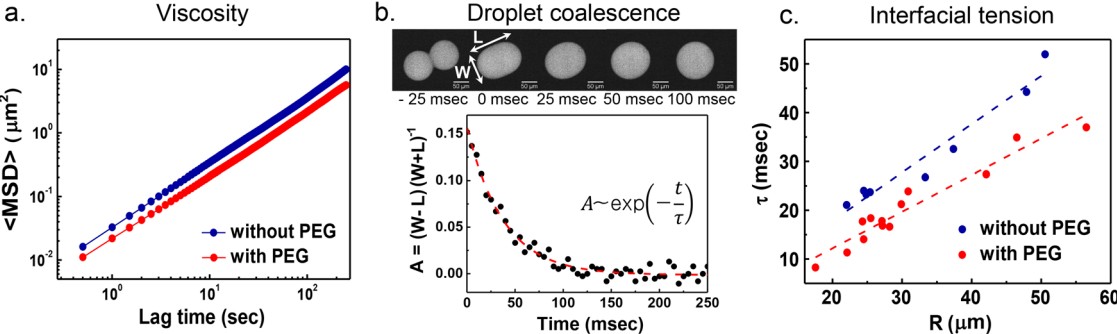

**Fig. 8 Viscosity and interfacial tension. a** MSD for tracer beads in coacervate with and without PEG. Viscosity was estimated from the Stokes−Einstein relation. **b** Coacervate droplet coalescence event in the presence of PEG. The coalescence event was well fit by an exponential decay, and this determined the relaxation time $t$. **c** The linear fit of the relaxation time with respect to the radius of the coacervate. The interfacial tension was calculated from the slope. Coacervate with and without PEG was formed at 0.1 M sodium acetate (pH 5.0), without additional NaCl.

interaction further diminishes the local densities of polyelectrolytes in the dilute region, which at high PEG concentration results in an increase in polyelectrolyte density in the dense phase. This is a remarkable result that PEG can induce a significant change in the phase behavior of polyelectrolytes at relatively strong electrostatic strength. The PEG density distribution is presented (Fig. 7d), and found to feature a binodal with two basins, in which the PEG-depleted basin corresponds to the dense region occupied by polyelectrolytes. In other words, PEG is mostly excluded from the dense CC region. These findings from FTS are in excellent agreement with conclusions derived from experimental results, namely that PEG as the neutral crowding agent predominantly resides in the dilute phase of coexistence, and that PEG enhances the driving force for CC by increasing the dehydration entropy. FTS furthermore revealed that a neutral crowder, such as PEG, can dramatically increase the density distribution of polyelectrolytes in the dense region.

**Viscosity and interfacial properties of εPL-HA CC**. We next turn to changes in viscosity or the interfacial tension of the coacervate induced by the addition of PEG. We utilized the microrheology technique to measure the viscosity of the coacervate formed with and without PEG. The mean squared displacement (MSD) of the probe particles followed a power law, MSD ~ $t^\alpha$ with lag time $t$, and the diffusive exponent $\alpha$ was near-unity (Fig. 8a). This exponent implies that the coacervate phase exhibited viscous characteristics within the explored timescales. The calculated viscosity was $2.44 \times 10^{-2}$ Pa s for the CC formed without PEG, and $4.06 \times 10^{-2}$ Pa s for CC formed with PEG. The addition of PEG almost doubled the coacervate viscosity, but this value is relatively low, considering the polyelectrolyte concentration in dense phase (without PEG: 188 mg mL$^{-1}$, with PEG: 321 mg mL$^{-1}$); note that the viscosity of water is $10^{-3}$ Pa s, and of honey (~700 mg mL$^{-1}$) is 10 Pa s. A relatively low viscosity may be meaningful to facilitate mass transfer for biomacromolecules to move into and out of the CC phase. Next, we estimated interfacial tension of the coacervate by exploiting coalescence events of the micro-sized coacervates. The relaxation time $\tau$ was calculated as the decay time scale of the aspect ratio of a deformed droplet under coalescence (Fig. 8b). Interfacial tension of the coacervate formed with and without PEG was calculated with the measured viscosity values obtained from microrheology. The determined interfacial tension of the coacervate formed with PEG ($5.15 \times 10^{-5}$ N m$^{-1}$) was about double that of coacervate formed without PEG ($2.34 \times 10^{-5}$ N m$^{-1}$) (Fig. 8c). These results show that PEG was able to stabilize the interface of coacervate droplets against bursting by increasing the

interfacial tension. While increased, both the viscosity and interfacial tension values were still relatively low considering the polyelectrolyte concentration inside the coacervate.

**Polyelectrolyte dynamics in εPL-HA CC with/without PEG**. We next used FRAP to study the exchange dynamics of fluorescence-labeled molecules. The coacervate suspension (containing FITC-εPL) was imaged with a fluorescence microscope. To monitor the diffusion of FITC-εPL in the coacervate droplet, its small region in the center was bleached (partial droplet bleaching). However, regardless of PEG, a bleached image could not be obtained, because the diffusion of εPL within the droplet was too fast (Fig. 9a). The diffusion of FITC-εPL was likely faster than the time capturing the frame by a camera (0.265 s per frame). This fast exchange is consistent with weak polyelectrolyte−polyelectrolyte and polyelectrolyte−water interactions in the coacervate droplet. The ODNP results also implied that the polyelectrolyte−water interaction was weak in the dense phase. Therefore, the FRAP and ODNP results show that polyelectrolytes are highly dynamic, and only weakly interact upon CC. This is consistent that bulk-like water is contained in the dense phase, and the constituents of the coacervate are freely diffusing within the coacervate domains[69].

What about the polyelectrolyte exchange between the coacervate droplet and the surrounding solution? We bleached the whole area of the single coacervate droplet formed with and without PEG (Fig. 9b). After bleaching, the fluorescence in the coacervate droplet formed without PEG gradually recovered over time. However, in the presence of PEG, the fluorescence hardly recovered for over 100 s (Fig. 9c), which implies that the polyelectrolyte exchange was immensely slowed down compared to exchange in the absence of PEG. This difference in the recovery might be due to the difference in the polyelectrolyte concentration of the surrounding solution, because the fluorescence recovery from the surrounding solution is dependent on the polyelectrolyte concentration in the surrounding solution[70]. Specifically, about 78% (w/w) of the polyelectrolytes (234 mg out of 300 mg initially added polyelectrolytes, Table 1) was in the macro-separated dilute phase formed in the absence of PEG, whereas about 33% (w/w) of the polyelectrolytes (98 mg out of 300 mg, Table 1) was in the dilute phase formed in the presence of 10% (w/v) PEG. The addition of PEG dramatically slowed down the polyelectrolytes exchange between the droplet and the surrounding solution, likely by depleting the dilute phase of polyelectrolyte constituents that can replenish the dense phase. This mechanism of polyelectrolyte condensation in the dense phase appears to stabilize the coacervate droplet and hinder constituent exchange.

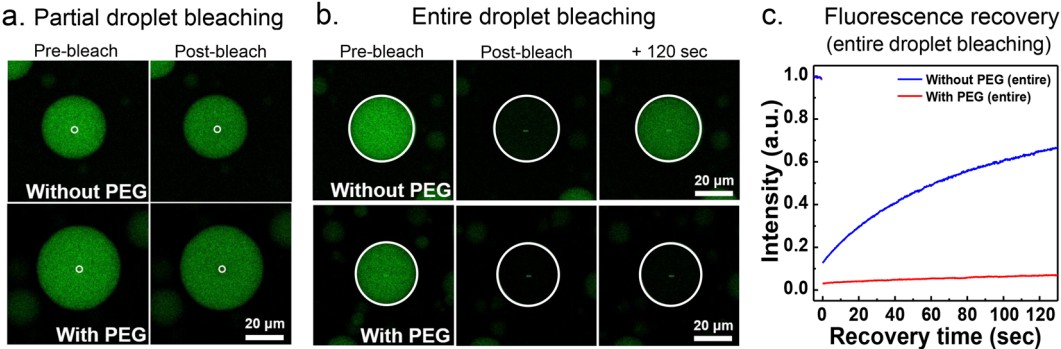

**Fig. 9 Fluorescence images of the εPL-HA coacervate droplets formed with and without PEG (0.1 M, pH 5.0 sodium acetate, without additional NaCl).** Two different bleaching geometries were used, and the white circles indicated the bleaching area. **a** In partial droplet bleaching (radius ~1.5 μm), droplet images represent the droplet before bleaching and right after bleaching. **b** In entire droplet bleaching (radius ~16 μm), droplet images represent the droplet before bleaching, right after bleaching, and after 120 s of bleaching. **c** The recovery curves after entire droplet bleaching. Blue: without PEG, Red: with PEG.

## Discussion

A series of experiments established that CC between εPL and HA is favored at lower salt concentration, in the presence of PEG, and at elevated temperatures. PEG, a molecular crowder, increased the coacervate volume fraction and density, but did not partition into the dense phase. Added PEG and elevated temperature stabilized the εPL-HA complex coacervates, even at high NaCl concentration of order 200 mM found under physiological condition. εPL-HA CC followed LCST behavior, although the polyelectrolytes had no hydrophobic constituents. We found that water−water, water−complex coacervate, and water−polyelectrolyte interactions are weak in εPL-HA CC. The CC constituents, εPL, HA, and surrounding water remained highly dynamic in the coacervate phase. The viscosity and interfacial tension of the coacervate droplets moderately increased upon addition of PEG, stabilizing εPL-HA CC. All results are consistent with the hypothesis that CC is driven by partial dehydration of the hydration and interstitial water, without polymer condensation. This is consistent with a process that would increase the total entropy upon CC, facilitated at elevated temperatures, in the presence of PEG, or both. Advanced FTS results of the density distribution for polyelectrolytes and PEG at coexistence are consistent with the key experimental finding that PEG drives CC by increased dehydration entropy, without partitioning into the coacervate phase. Our study does not exclude the potential role of entropy gain from counterion-release—the currently widely accepted hypothesis—as an additional driver of CC. Rather, our study shows that entropy gain from water-release is a major contributor for CC that can be further amplified in the presence of highly hydrophilic and neutral crowders such as PEG, and can be sufficient to reproduce entropy-driven CC. However, our study did not examine nor directly compare the effect of counterion-release as an additional factor.

## Methods

**Materials.** ε-poly-L-lysine (4 kDa) was purchased from Shinseung Hichem (Seoul, Korea). Hyaluronic acid (5 kDa) was purchased from Bioland (Seoul, Korea). Poly (ethylene glycol) (10 kDa), 4-maleimido-TEMPO (4MT) were purchased from Sigma-Aldrich (St. Louis, USA). 6-(Fluorescein-5-(and-6)-Carboxamido) Hexanoic Acid, Succinimidyl Ester (SFX) was purchased from Thermo Fisher (San Jose, USA). Spin-labeled εPL was obtained by adding double excess of 4MT to εPL solution. FITC-labeled εPL was obtained by adding four times excess of SFX to εPL solution.

**Preparation of εPL-HA complex coacervates.** εPL-HA complex coacervates formed with and without PEG were prepared. εPL (10 mg mL$^{-1}$) and HA (10 mg mL$^{-1}$) were prepared in sodium acetate buffer (0.1 M, pH 5.0). Then the microdroplet coacervate suspension was prepared by mixing εPL solution and HA solution, and the yield of the dense CC phase was qualitatively assessed by

relative turbidity at this stage. The relative turbidity was found to be maximized at a mass ratio of 2 (εPL):8 (HA) (Supplementary Fig. 2). This ratio corresponds to a mixture where net charge neutralization of εPL and HA is expected, and henceforth all coacervate samples were prepared at a mass ratio of 2:8. In some cases, 10% (w/v) of PEG or certain concentration of NaCl was additionally added to the solution of εPL and HA. Immediately after the solutions had been mixed, optical microscope images of coacervate suspension with and without PEG at additional 40 mM NaCl were captured. The turbidity of coacervate suspension was measured at 500 nm at a range of NaCl concentrations (0−200 mM), consistently within 5 s of mixing of the HA and εPL solution. As the relative turbidity also depends on the microdroplet size, and hence changes with time as the microdroplets coalesce, a more accurate quantification method of the coacervate phase was required for further study. For this purpose, the microdroplet coacervate suspension was separated to macro-separated dilute and dense phases by centrifugation (4876 × g, 10 min, RT). The volume fraction of the macrophase-separated coacervate was evaluated as $V_{dense}/V_{total}$. The density of the dense coacervate phase was determined by weighing the specific volume of the coacervate phase after macrophase separation. The polyelectrolyte concentration in the macro-separated dense coacervate and dilute phase was estimated as (the weight of the polyelectrolytes)/(the volume of the macro-separated dense or dilute phase). The weight of the polyelectrolytes in each phase was calculated, and the volume of the macro-separated dense and dilute phase was measured by pipetting (details are described in Supplementary Table 1).

**Temperature study of εPL-HA complex coacervation.** Temperature effect on εPL-HA complex coacervation was measured by capturing optical microscope images of the coacervate suspension at specific temperatures, which were set using a temperature controller. The absorbance of the coacervate suspension with different PEG% (w/v) was scanned from 25 to 80 °C. The absorbance of the suspension was recorded at 25, 70, and 90 °C for 10 min. Absorbance measurements were performed using a temperature-controllable circular dichroism spectro-polarimeter (J-815, Jasco, Japan).

**Pulsed-field gradient NMR.** Pulsed-field gradient NMR measurements were performed on a Bruker Avance III Super WB spectrometer equipped with a Bruker DIFF50 diffusion probe with replaceable RF inserts, and the diffusion probe was tuned to $^1$H nuclei. $^1$H diffusion measurements were performed on the macrophase-separated samples at 25 °C. Samples were equilibrated at that temperature for 15 min before measurement. A pulse sequence of stimulated echoes with bipolar pulses was used to measure diffusion coefficients. The attenuation of the echo E was fit to $E = \exp(-(\gamma g \delta)^2 D(\Delta - \delta/3))$, where $\gamma$ [s$^{-1}$ G$^{-1}$] is the gyromagnetic ratio, $g$ [G cm] is the gradient strength, $\delta = 1$ ms is the duration of the gradient pulse, $\Delta = 20$ ms is the interval between gradient pulses, and $D$ [m$^2$ s$^{-1}$] is the diffusion coefficient. For each diffusion measurement, 16 experiments were performed at various $g$. All measured attenuations were adequately fit with single-exponential decays.

**Electron paramagnetic resonance.** Electron paramagnetic resonance experiments were performed with an X-band (0.35 T) Bruker EMXPlus spectrometer using a high sensitivity microwave cavity of Bruker ER 4119HS-LC (Bruker, Massachusetts, USA). Samples (3.5 μL) were put into quartz capillaries of 0.6 mm ID × 0.84 mm OD (Vitrocom, New Jersey, USA), and both ends were sealed with Critoseal. Then, they were placed into 4-mm diameter open-end quartz EPR tubes.

**Overhauser dynamic nuclear polarization.** Overhauser dynamic nuclear polarization was performed on samples (3.5 μL) that contained spin-labeled εPL. The

samples were loaded into quartz capillaries of 0.6 mm ID × 0.84 mm OD (Vitrocom, New Jersey, USA), and both ends of the tubes were sealed with Critoseal. ODNP experiments were performed using a Bruker EMXPlus spectrometer and a Bruker Avance III NMR console (Bruker, Massachusetts, USA). The capillary tube was mounted on a home-built NMR probe with a U-shaped NMR coil, and was set in a Bruker ER 4119HS-LC sensitivity cavity. Samples were irradiated at 9.8 GHz with the center field set at 3484 G and sweep width of 120 G. Dry air was streamed through the NMR probe during all measurements. Theory of ODNP and details in the experiment are previously reported in other studies[65–67].

**Moisture analyzer**. After macroscopic LLPS, the water content of the macro-separated dense coacervate phase was measured by moisture analyzer (MB35, OHAUS, New Jersey, USA). Specific volume (>0.5 g) of the dense coacervate phase that had been formed with PEG or without PEG was dropped to the inner dish of the analyzer. The analyzing temperature was set to 110 °C.

**Microrheology**. Microrheology measurements were performed with εPL-HA coacervates with and without PEG. To visualize the coacervate droplet, 1% (w/v) of FTIC-labeled εPL was contained in the εPL solution (10 mg mL⁻¹). After thoroughly vortex-mixed, the samples were centrifuged (3000 × g, 1 h) and relaxed in dark for 4 h at room temperature. A clear macro-scale separation of the dilute and dense coacervate phases upon excitation of FITC was observed on a trans-illuminator, and ~5 μL of dense coacervate phase for each sample was obtained using a micropipette. Fluorescent carboxylate-modified polystyrene particles ($d = 2$ μm) was added in the samples and thoroughly pipet-mixed. The mixtures were then introduced into coverslip-sandwiched fluid chambers and subsequently sealed with 5-min epoxy. Confocal microscopy was performed at 561 nm excitation, where the polystyrene particles in the coacervate phase were imaged every 500 ms. Further particle-tracking analyses were performed using MATLAB software (Mathworks, Massachusetts, USA) to obtain MSD of the particles diffusing in εPL-HA coacervate droplet formed with and without PEG. The particle diffusion coefficients $D$ are calculated from $\langle MSD \rangle = 4Dt$ and used to calculate the viscosity ($\eta$) of the coacervate phase via the Stokes−Einstein relation, $D = k_B T (6\pi\eta r)^{-1}$, where $k_B$ is Boltzmann's constant, $T$ is temperature, and $r$ is the probe radius (1 μm).

**Interfacial energy measurement by droplet coalescence**. The dynamics of the coalescence process of two spherical coacervate droplets were studied to measure the interfacial tension of the εPL-HA coacervates formed with and without PEG. The coacervate suspensions were rotate-incubated at room temperature for 2 h to obtain desired sizes of the εPL-HA coacervate droplets for confocal imaging. To minimize friction from the surface during droplet coalescence events, we utilized a flat oil/water interface where the coacervate droplets can diffuse laterally on a surfactant-stabilized oil/water interface in a coverslip-sandwiched fluid chamber[71]. After the samples were prepared, coalescence events were imaged on a ×10 objective with confocal microscopy every 5 ms. The interfacial tension $\gamma$ of the dense coacervate phase was determined from the time scale of the progress of the relaxation via Eq. 1.

$$\tau \cong \frac{19}{20} \frac{\eta R}{\gamma}, \tag{1}$$

where $\tau$ is the decay time of $A$, a ratio of the difference of the length and width and the sum of the length and width of a deformed droplet under coalescence, $\eta$ is viscosity of the coacervate phase, and $R$ is the droplet radius after the coalescence[72]. We followed previously reported experimental methods of both microrheology and droplet coalescence[73].

**Fluorescent recovery after photobleaching**. Fluorescence recovery after photobleaching experiments were conducted using a confocal microscope (Leica TCS SPX, Leica, Germany) with a suspension of εPL-HA coacervate that included <1% (w/v) FITC-εPL. For this measurement, a ×10 DRY objective was used. A 488-nm laser was used to excite the FITC. Images of 256 × 256 pixels were acquired at exposure times of 0.265 s per frame. Similar with previous studies[70], partial droplet bleaching was used to understand the diffusion of FITC-εPL within the droplet, and entire droplet bleaching was used to know the exchange of FITC-εPL between the inside and the outside of the droplet. Bleaching was performed with 100% laser power with either partial droplets or entire droplets. The region of interest was the volume within 1.5 μm radius of the center for partial droplet bleaching, and the entire droplet ($D \sim 30$ μm) for whole-droplet bleaching.

## Data availability

The datasets generated during the current study are available in the Figshare repository, https://doi.org/10.6084/m9.figshare.12097812.v1.

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

## Acknowledgements

The work of B.-j.J. was supported by the U.S. Department of Energy (DOE), Office of Science, Basic Energy Sciences (BES), under Award # DESC0014427. D.S.H. was supported by the Basic Science Research Program through the National Research Foundation of Korea (NRF) funded by the Ministry of Science and ICT (NRF-2016M1A5A1027592, NRF-2016M1A5A1027594, and NRF-2017R1A2B3006354). Support for the ODNP studies was provided by the Deutsche Forschungsgemeinschaft (DFG, German Research Foundation) under Germany's Excellence Strategy—EXC-2033—Project number 390677874. Studies of LLPS by S.H. and J.-E.S. were supported by the National Institutes of Health (NIH) under Grant Number R01AG05605, while computational method development for CC by J.-E.S. and G.H.F. was supported by the MRSEC Program of the National Science Foundation under Award No. DMR 1720256. J.-E.S. acknowledges support from the National Science Foundation NSF under Award No. MCB-1716956 for the CC simulations. FTS used resources of the Extreme Science and Engineering Discovery Environment (XSEDE, supported by the NSF Project TG-MCA05S027) and the Center for Scientific Computing from the California NanoSystems Institute UC Santa Barbara (CNSI) available through the Materials Research Laboratory (MRL): an NSF MRSEC (DMR-1720256) and NSF CNS-1725797.

## Author contributions

S.P. and B.-j.J. performed the experiments. R.B. and Y.L. helped with experiments. S.H. and D.S.H. supervised the project and contributed to experimental design. G.H.F., K.T.D., and J.-E.S. planned and supervised the simulations and S.N. performed them. S.P., B.-j.J., S.H. and D.S.H. wrote the manuscript and G.H.F., S.N., K.T.D., and J.-E.S. edited the manuscript.

## Competing interests

The authors declare no competing interests.
