## [Peer Review File · Communications Chemistry]

Reviewers' comments:

Reviewer #1 (Remarks to the Author):

The experimental results in this paper are impeccable and certainly deserve to be published. The authors demonstrate the PEG promotes and densifies the dense phase by crowding and dehydration, without physically entering the the dense phase itself. These observations are original and important. However, there are many points, especially of the introductory discussion to which I object and find misleading. First of all, counterion release, not mentioned, is often the greatest entropic contribution to CC formation. Secondly, I do not accept the characterization of this CC pair as simple. Poly- ϵ -lysine introduces a significant stretch of hydrophobicity in the backbone; hyaluronic acid is a more straightforward structure as a linear, hydrophilic polyelectrolyte but has an indisputably complicated backbone. I am not certain that the hydrophobicity of poly- ϵ -lysine is not responsible for the LCST type behavior observed. There is more phase behavior published than this paper gives credit to. A truly simple PEC system is polyK-polyE for which a rather complete phase diagram was published in *Macromolecules* last year(51, 2988-2995). It is a shame that these authors did not really produce a phase diagram. This paper should be published when these comments are properly addressed. One final comment, the first word of the paper, "Whilst" is not a modern American English word. Shakespeare, maybe; sounds anachronistic.

Reviewer #2 (Remarks to the Author):

See document in attachment.

Reviewer #3 (Remarks to the Author):

In this work, the authors investigate complex coacervation (CC) of a simplified system comprising poly-L-lysine and hyaluronic acid. Using a combination of turbidity measurements, microscopy, NMR PFG diffusion, ODNP, FRAP and micro-rheology the influence of temperature, ionic strength and concentration of the molecular crowder PEG is investigated. The combined data support the hypothesis that CC of this simple system is mainly driven by entropy. Because the study is done in a very systematic manner, it will be highly useful for other researchers interested in liquid-liquid phase separation. The manuscript is well written and the data are presented in appropriate figures. This is a nice piece of work.

Minor point: In none of the figure legends it is described what the error bars represent.

Dear Editor,

With interest I have read the article on “Dehydration entropy drives liquid-liquid phase separation by molecular crowding” by Park et al.

The work presented is very interesting, but in my view there is another explanation for the observed phenomena. PEG is used as a molecular crowder and its effect on the complex coacervate (CC) formation between Hyaluronic Acid (HA) and polylysine (PLL) is studied. PEG is widely used as a molecular crowding agent, but this polymer is also known to segregatively phase separate with other polymers, e.g., Dextran. As a result two phases are formed, one rich in dextran one rich PEG. In these systems it could be that mixing them at equal mass concentration does not result in the two phases having equal volumes. For instance gelatin and dextran need to be mixed at a mass ratio of 6:4 in order to get the same volumes of the phases, indicating that by segregative phase separation the water can be distributed differently over the phases, which also seems to be the case in the PEG/PLL+HA system. Further it might be that salt ions have a preference for either of the phases (also for gelatin and dextran), so the density of these phases might be dependent on the ionic strength.

I think that this system is a mixture of associative (PLL+HA) and segregative (PEG/PLL+HA) phase separation and that the results can be explained accordingly. Topic wise I do think that this work may contribute to a better understanding of Liquid-Liquid phase separation I therefore recommend major revisions of the article.

Other points

- Line 54: I suggest to change the sentence to CC is a phenomenon in which oppositely charged polyelectrolytes separate into...
- Line 57: Weakly interacting polyelectrolytes is confusing, probably it is meant that charges on the polyelectrolytes are screened to a certain extent? Be careful by using weakly in the same sentence of polyelectrolytes (also other places within the document). Weak polyelectrolytes are a certain class of polyelectrolytes of which the charge is pH-dependent.
- Line 64: please include references especially to the one where crowding pressure is important. What is the difference with your system?
- Line 68: Cohen Stuart (doi.org/10.1016/j.jcis.2011.05.080 and doi.org/10.1021/jp809489f) et al have sketched ΔG , ΔH and ΔS as function of the ionic strength, you may want to refer to this sketch and be more specific here.
- Line 87: What is meant by isotonic conditions?
- Line 116: Why do you use pH 5? Is this the optimal pH at which CC occurs? It seems rather low, HA will not be fully charged at this pH.

Line 122: What assumptions did you make to calculate this ratio? Also would it be possible to convert the x-axis of FIG S2 to charge ratio?

Line 119: What about the kinetics of phase separation? In most of these systems the turbidity instantly increases upon mixing. Did you use a stopped-flow set-up to measure the turbidity?

Figure 1b: You measure the effect of ionic strength on the phase separation, but only discuss your results with PEG, not without PEG. I find your results really interesting, your droplets disintegrate at 200 mM NaCl without PEG and remain intact when PEG is present. Do the latter dissolve above a certain ionic strength? Do ions partition differently over the two phases?

Line 208: Here you state that you have a micro separated system, in line 211 you state that you haven't. Please change the text or explain better what is meant.

Reviewer: 1

Comments:

The experimental results in this paper are impeccable and certainly deserve to be published. The authors demonstrate the PEG promotes and densifies the dense phase by crowding and dehydration, without physically entering the dense phase itself. These observations are original and important. However, there are many points especially of the introductory discussion to which I object and find misleading.

First of all, counterion release, not mentioned, is often the greatest entropic contribution to CC formation.

We appreciate the reviewer for the comment about the importance of counterion-release. We agree that the entropy gain of counterion releasing has been proposed to be the main driving forces for complex coacervation (CC) of oppositely charged polyelectrolytes. However, a very recent paper by our group and collaborators [1] demonstrated that CC, and even entropy-driven CC displaying LCST (lower critical solution temperature) behavior, can be modeled without at all invoking counterion release. Instead, this paper and other recent publications [2] proposed that hydration water release from the polymer surface due to hydration shell sharing upon polymer association could be a major contributor for entropy gain, more so than counterion-release. We are not arguing that counter ion release cannot occur or cannot be responsible for entropy gain. Rather, we show that CC and LCST processes do not rely on or need counterion release to robustly occur. We now briefly discuss the topic of counter ion release explicitly. Example quotes from revised manuscript include:

Introduction: “The origin of the entropy gain is assumed in the literature to be due to counterion-release.³⁵⁻³⁸ However, a recent study relying on experimental and field-theoretic simulations (FTS) of CC between the IDP tau and RNA demonstrated that counterion-release to be a negligible driver of CC, or at least does not need to be invoked to replicate LCST driven CC,²² while hydration water-release may be a major contributor to entropy gain. In a study by Cohen Stuart, counterion-release entropy was found to be most negative at the lowest ionic strength.³⁷ However, it is important to note that both counterion-release and dehydration entropy would be greater at lower ionic strength where the effective surface charge of the polyelectrolytes is greater.”

Discussion: “Counterion-release has been a long-standing hypothesis for CC formation. Our study shows that entropy gain from water-release is a major contributor for CC formation, can be amplified in the presence of highly hydrophilic and neutral crowders such as PEG, and can be sufficient to reproduce entropy driven CC. However, our study does not examine nor exclude counterion release as an additional factor.”

Ref

[1] Lin, Y., McCarty, J., Rauch, J. N., Delaney, K. T., Kosik, K. S., Fredrickson, G. H., ... & Han, S. (2019). Narrow equilibrium window for complex coacervation of tau and RNA under cellular conditions. *Elife*, 8, e42571.

[2] Ribeiro, S. S., Samanta, N., Ebbinghaus, S., & Marcos, J. C. (2019). The synergic effect of water and biomolecules in intracellular phase separation. *Nature Reviews Chemistry*, 3(9), 552-561.

Secondly, I do not accept the characterization of this CC pair as simple. Poly-ε-lysine introduces a significant stretch of hydrophobicity in the backbone; hyaluronic acid is a more straightforward structure as a linear, hydrophilic polyelectrolyte but has an indisputably complicated backbone. I am not certain that the hydrophobicity of poly-ε-lysine is not responsible for the LCST type behavior observed. There is more phase behavior published than this paper gives credit to. A truly simple PEC system is polyK-polyE for which a rather complete phase diagram was published in Macromolecules last year (51, 2988-2995). It is a shame that these authors did not really produce a phase diagram. This paper should be published when these comments are properly addressed.

Regarding the simplicity argument for the HA-εPL system:

What we meant to convey with the descriptor “simple” in our paper is that our HA-εPL system is lacking sequence complexity. However, as far as we know, this reviewer seems to point out that our εPL has a different structure compared to poly-L-lysine (PLL) (see as follows; left: εPL, right: PLL).

We do not know how much this structural difference can affect the basic behavior of CC, specifically as a function of added PEG, but we can certainly say that εPL and PLL have no sequence complexity compared to protein-based polyelectrolytes. In the paper the reviewer has listed (<https://pubs.acs.org/doi/10.1021/acs.macromol.8b00238>) the authors studied the CC between poly-L-lysine (polyK) and poly-glutamic acid (polyE), and indeed these two polyelectrolytes have identical hydrophilic backbones. However, whether studying CC of oppositely charged polyelectrolytes with identical or different backbone identity renders a more general or less general system is unclear. However, once we state in the manuscript that our HA-εPL system was designed to have no sequence complexity and minimal hydrophobicity, and that in comparison to protein-based polyelectrolyte systems, we hope we have resolved the misunderstanding. We now clarify this aspect as follows:

Abstract: “Significantly, entropy-driven LLPS was recapitulated with charged polymers lacking hydrophobicity and sequence complexity, and its propensity dramatically enhanced by PEG.”

Introduction: “The CC between polyelectrolytes with minimal sequence complexity and hydrophobicity compared to protein will teach us about the base property of CC, especially under conditions that mimic the cellular environment.”

The comment related on the phase diagram:

The reviewer is referring to the generation of a phase diagram as follows that relied on MD simulations and Thermogravimetric Analysis (TGA) of many macro-phase separated solutions that give the weight information of water, polymer, and salts in solution.

Generating this kind of phase diagram with our HA-PLL CC with/without PEG, even for a minimal number of parameters is extremely resource consuming, as (1) we need to macroscopically phase separate and generate both dilute and dense phases, at (2) many different NaCl conditions, as well as (3) with and without PEG, constituting 30-40 samples, while each TGA analysis would take about 5 hours per sample. More importantly, such study may not yield in-depth mechanistic and physical insight we are looking for, beyond what we have already demonstrated—namely proof of principle trends of the effects of PEG on CC, unless we join forces with computational experts to comprehensively model the effect of PEG on CC such that generalizable knowledge can be generated. This is exactly what we have done now. We launched an entirely new study, in collaboration with Profs. Glenn Fredrickson and Joan Shea, together with their two Postdocs, to perform field theoretic simulations (FTS) of CC of a three-component system—a first—constituted of a polycation, polyanion and PEG, with polymer size, excluded volume, charge density and interaction properties modeled adequately. Since this was the first time a FTS of three-component CC was performed with mixed charged and neutral polymers, method developments had to first ensue before the experimental CC system could be simulated. This took time, which is why our rebuttal effort was delayed, but this strategy paid off. Four co-authors were added, a main Figure 9 added, together with additional text on page 18-20. The method and finding is as follows:

We performed FTS using Complex Langevin sampling to characterize the influence of PEG on the liquid-liquid phase separation by CC of oppositely charged polyelectrolytes by using a coarse-grained bead-spring model of polymers to model HA and εPLL. Specifically, we determined the relative density of the polymer chains at the two-phase coexistence state, where the supernatant and the dense regions coexist. As a proxy of the dehydration propensity of polyelectrolytes, we modulated the mutual excluded volume interaction term and determined the density state of the complex coacervated polymers. To our delight we observed that the presence of PEG substantially enhanced the coacervation propensity by amplifying the density of the polyelectrolytes in the dilute and dense domains of coexistence, with PEG not or minimally partitioning into the dense coacervate phase, in agreement with experimental results. This exciting new result is now included as a highlight of our revised manuscript with Figure 9 and description on page 18-20.

Figure 9. Panel I-a and I-b show the density distribution

n of polyelectrolytes and PEG, respectively, in a two-phase solution in which a dilute supernatant and dense coacervate phase of polyelectrolytes coexist. Panel II-a and II-b show the density distribution (normalized histogram) of polyelectrolytes and PEG, respectively. In this system, we set the total density of polyelectrolytes and PEG at $\rho_P b^3 = 1.5$ and $\rho_C b^3 = 15$. The black, orange and blue curves show the density distribution of polyelectrolytes without PEG and with PEG at cross-excluded volume strengths $v_{PC} = 0.034 b^3$ and $0.068 b^3$, respectively.

One final comment, the first word of the paper, "Whilst" is not a modern American English word. Shakespeare, maybe; sounds anachronistic.

We appreciate to point this out. We corrected this "While".

Reviewer: 2

Comments

With interest I have read the article on "Dehydration entropy drives liquid-liquid phase separation by molecular crowding" by Park et al.

The work presented is very interesting, but in my view there is another explanation for the observed phenomena. PEG is used as a molecular crowder and its effect on the complex coacervate (CC) formation between Hyaluronic Acid (HA) and polylysine (PLL) is studied. PEG is widely used as a molecular crowding agent, but this polymer is also known to segregatively phase separate with other polymers, e.g., Dextran. As a result two phases are formed, one rich in dextran one rich PEG. In these systems it could be that mixing them at equal mass concentration does not result in the two phases having equal volumes. For instance gelatin and dextran need to be mixed at a mass ratio of 6:4 in order to get the same volumes of the phases, indicating that by segregative phase separation the water can be distributed differently over the phases, which also seems to be the case in the PEG/PLL+HA system.

Further it might be that salt ions have a preference for either of the phases (also for gelatin and dextran), so the density of these phases might be dependent on the ionic strength.

I think that this system is a mixture of associative (PLL+HA) and segregative (PEG/PLL+HA) phase separation and that the results can be explained accordingly. Topic wise I do think that this work may contribute to a better understanding of Liquid-Liquid phase separation I therefore recommend major revisions of the article.

We appreciate the reviewer for introducing us to concepts we had missed before, suggesting that this system of HA-PLL complex coacervation in the presence of PEG may be due to a mixture of mechanisms following associative (PLL+HA) and segregative phase separation (between PEG and PLL/HA). According to the reported papers [1], segregative phase separation is likely promoted by repulsion between different polymers, e.g. in Gelatin – Dextran [2], and the affinity of the polymer to the solvent. In our system, HA-PLL complexes and PEG do not interact with each other, and have highly different solvent affinity (PEG has a much higher affinity to water), hence PEG and HA/PLL complexes experience repulsive interactions. At the same time, PEG drives dehydration of HA and PLL, and promotes their association. Hence, our system could be viewed as a mixture of associative (HA+PLL) and segregative phase separation (PEG and HA/PLL). We add now following wording to respond to Reviewer 2:

“In another view, ϵ PL-HA CC with PEG may promote a mixture of associative and segregative phase separation.⁵⁹ Associative phase separation, e.g. complex coacervation, is driven by attraction between the biopolymers, while segregative phase separation, e.g. gelatin/dextran,⁶⁰ is promoted by an effective repulsion between the biopolymers. Based on these concepts, ϵ PL-HA complex coacervation can be interpreted as being promoted by associative phase separation driven by attraction of ϵ PL and HA, and the exclusion of PEG due to segregative phase separation between ϵ PL-HA complexes and PEG.”

Ref)

[1] McClements, David Julian. "Non-covalent interactions between proteins and polysaccharides." *Biotechnology advances* 24.6 (2006): 621-625.

[2] Edelman, Marijke W., et al. "Compatibility of gelatin and dextran in aqueous solution." *Biomacromolecules* 2.4 (2001): 1148-1154.

Other points

Line 54: I suggest to change the sentence to CC is a phenomenon in which oppositely charged polyelectrolytes separate into...

We thank the reviewer for the suggestion. We have revised the statement as follows:

“CC is a phenomenon in which polyelectrolytes separate into a dense polyelectrolyte-rich phase (coacervate phase) and a polyelectrolyte-depleted phase (dilute phase).^{23,}”

Line 57: Weakly interacting polyelectrolytes is confusing, probably it is meant that charges on the polyelectrolytes are screened to a certain extent? Be careful by using weakly in the same sentence of polyelectrolytes (also other places within the document). Weak polyelectrolytes are a certain class of polyelectrolytes of which the charge is pH dependent.

We clarified the statement as follows:

“CC typically occurs when oppositely-charged polyelectrolytes (referring to either the entire biopolymer or a biopolymer segment) interact with each other by electrostatic attraction, and ultimately form polyelectrolyte microdroplets termed the complex coacervate phase.”

Line 64: please include references especially to the one where crowding pressure is important.

What is the difference with your system?

Cellular environment contains high concentration of macromolecules. To understand LLPS involving proteins in the cellular environment, molecular crowder are used to mimic the cellular environment. [1] The literature suggests that crowding agents lead to increased effective concentration of solutes, which can lower the reaction rate of association. [2] However, in this paper, we discovered that the role of PEG on CC is dehydrating the polymer constituents of CC (HA and PLL), and suggest that this dehydration contributes to an increased entropy.

We added the following references on the crowding effect on CC as follows:

Ref) [1] Marianelli, A. M., B. M. Miller, and Christine Dolan Keating. "Impact of macromolecular crowding on RNA/spermine complex coacervation and oligonucleotide compartmentalization." *Soft matter* 14.3 (2018): 368-378.

[2] Kim, Jun Soo, and Arun Yethiraj. "Effect of macromolecular crowding on reaction rates: a computational and theoretical study." *Biophysical journal* 96.4 (2009): 1333-1340.

We have now also added the reference on the crowding effect on CC as follows:

"CC is affected by many factors including ionic strength, pH, polyelectrolyte concentration, a balanced mixing ratio of oppositely-charged polyelectrolytes, molecular weight of the polyelectrolytes, as well as temperature and the crowding pressure³⁰."

Ref)

[30] Kim, Jun Soo, and Arun Yethiraj. "Effect of macromolecular crowding on reaction rates: a computational and theoretical study." *Biophysical journal* 96.4 (2009): 1333-1340.

Line 68: Cohen Stuart (doi.org/10.1016/j.jcis.2011.05.080 and doi.org/10.1021/jp809489f) et al have sketched ΔG , ΔH and ΔS as function of the ionic strength, you may want to refer to this sketch and be more specific here.

We now refer to this important study in the revised manuscript. However, we also explain why the results in the Cohen Stuart paper does not contradict our proposed mechanism of entropy gain via dehydration. Some of the important papers were unintentionally left, so we have cited these in the revised manuscript with a brief explanation as follows:

"The origin of the entropy gain is assumed in the literature to be due to counterion-release.³⁵⁻³⁸ However, a recent study relying on experimental and field-theoretic simulations (FTS) of CC between the IDP tau and RNA demonstrated that counterion-release to be a negligible driver of CC, or at least does not need to be invoked to replicate LCST driven CC,²² while hydration water-release may be a major contributor to entropy gain. In a study by Cohen Stuart, counterion-release entropy was found to be most negative at the lowest ionic strength.³⁷ However, it is important to note that both counterion-release and dehydration

entropy would be greater at lower ionic strength where the effective surface charge of the polyelectrolytes is greater.”

Line 87: What is meant by isotonic conditions?

We intended “isotonic conditions” as “cellular conditions”, but the term doesn’t seem to be clear. We corrected the sentence for better understanding as follows:

“We have empirical evidence that the effect of crowding and excluded volume effects are key factors that stabilize the CC of IDPs under cellular conditions.”

Line 116: Why do you use pH 5? Is this the optimal pH at which CC occurs? It seems rather low, HA will not be fully charged at this pH.

The pKa of the ϵ PL amine groups is close to 10, and the pKa of HA is about 3. We agree with the reviewer that CC could be more favorable in slightly higher pH than at pH 5 that was chosen as our experimental condition. However, CC is considered the concept of how cells might concentrate proteins in secretory granules and secrete the protein to the environment. We chose pH 5, as the internal pH of reported secretory granules is in the range of pH 5 – 6. [1, 2, 3,4]

Ref)

[1] Waite, J. Herbert, et al. "Mussel adhesion: finding the tricks worth mimicking." The journal of adhesion 81.3-4 (2005): 297-317.

[2] De Jong, HG Bungenberg, and H. R. Kruyt. "Colloid science." R. Kruyt (ed.), Elsevier, Amsterdam (1949), Vol. II, Chapter 11, 433-482.

[3] Casey, Robert P., et al. "Active proton uptake by chromaffin granules: observation by amine distribution and phosphorus-31 nuclear magnetic resonance techniques." Biochemistry 16.5 (1977): 972-977.

[4] Hutton, John C. "The internal pH and membrane potential of the insulin-secretory granule." Biochemical Journal 204.1 (1982): 171-178.

Line 122: What assumptions did you make to calculate this ratio? Also would it be possible to convert the x-axis of FIG S2 to charge ratio?

A maximum turbidity point represents the yield and the highest number density of CC droplets, especially as the CC leads to the formation of micron-sized droplets that effectively scatter light at $\lambda = 500$ nm, and hence represent the conditions under which net charge neutralization of PLL and HA is achieved. Given that we didn’t directly measure the exact charge of polyelectrolytes, this turbidity result only can estimate the charge ratio rather than the exact value.

Line 119: What about the kinetics of phase separation? In most of these systems the turbidity instantly increases upon mixing. Did you use a stopped-flow set-up to measure the turbidity?

As the reviewer pointed out, the turbidity of the CC changes over time. Given that this ripening time can be very long, it is challenging and practically infeasible to wait for

complete macro-phase separation. We remedied this dilemma by measuring the turbidity of the sample under the same preparation conditions and time upon CC formation to conduct comparisons between different samples. Specifically, we checked the turbidity right after the 5sec mixing of the HA and ϵ PL solution.

Figure 1b: You measure the effect of ionic strength on the phase separation, but only discuss your results with PEG, not without PEG. I find your results really interesting, your droplets disintegrate at 200 mM NaCl without PEG and remain intact when PEG is present. Do the latter dissolve above a certain ionic strength? Do ions partition differently over the two phases?

We additionally checked CC at 300 mM and 400 mM NaCl in the presence of PEG, but we firmly confirmed that CC did not occur at 400 mM NaCl even in the presence of PEG. About the ion partitioning, we did not measure the ion concentration of each phase. However, in the presence of PEG, CC occurs even at a high concentration of NaCl, so we estimate that PEG has the ability to hold a certain amount of NaCl and stay in the dilute phase.

Line 208: Here you state that you have a micro separated system, in line 211 you state that you haven't. Please change the text or explain better what is meant.

We revised this misleading statement as follows:

“a mixed solution of ϵ PL and HA (with additional 60 mM NaCl) in the presence and absence of PEG was placed on a temperature-controlled stage of an optical microscope”

Reviewer: 3

Comments:

In this work, the authors investigate complex coacervation (CC) of a simplified system comprising poly-L-lysine and hyaluronic acid. Using a combination of turbidity measurements, microscopy, NMR PFG diffusion, ODNP, FRAP and micro-rheology the influence of temperature, ionic strength and concentration of the molecular crowder PEG is investigated. The combined data support the hypothesis that CC of this simple system is mainly driven by entropy. Because the study is done in a very systematic manner, it will be highly useful for other researchers interested in liquid-liquid phase separation. The manuscript is well written and the data are presented in appropriate figures. This is a nice piece of work.

Minor point: In none of the figure legends it is described what the error bars represent.

We appreciate the reviewer's positive comments. We added the missing descriptions into the legend of Figure 2:

“All values expressed as (mean \pm standard deviation).”

And Figure 7:

“Error bars indicate the standard deviation.” and “Error bars with the 5% deviation on the means have been marked to show the effect of experimental uncertainty.”

REVIEWERS' COMMENTS:

Reviewer #1 (Remarks to the Author):

This is a significantly enhanced manuscript, especially with the addition of the theoretical calculations that support the overall interpretation of the experiments. I remain unsatisfied with the assertion that hydrophobicity plays no role in these experiments as there is a significant hydrophobic portion of the polycation, and therefore, I believe the multiple further assertions that counterion release is not important in complex coacervation are wrong and misleading. This system is not "lacking hydrophobicity" as they claim in the abstract and later. I do accept that in the system under study here the authors have made a convincing case that dehydration is the source of entropy gain. Their multiple assertions essentially claim an unwarranted generality of their results. If the authors would tone this down I would have no objection to publication. A final picky point ... I appreciate the change from "Whilst" to "While" in the opening line, but that opening line is not a complete sentence. I think they should just drop "While" and start "Fifty percent of protein sequences ... "

Reviewer #2 (Remarks to the Author):

I am very happy about the improved manuscript and in my opinion I think all the issues have been resolved and comments have been addressed.

My recommendation: publish as it is.

Reviewer #3 (Remarks to the Author):

The authors have addressed my minor suggestions.

REBUTTAL:

In response to the Editor's comment: "Reviewer #1 remains unsatisfied by the claims for generality in your manuscript, but we feel these concerns can be addressed by making it clear in the discussion that, in general, counterion release is important in complex coacervates. We are confident that we should be able to assess the response in-house but we reserve the right to contact the reviewer again if we do not think that these requests have been fully addressed."

We revised our conclusion as follows, in which we make clear that our study does not exclude the possibility that counterion release plays an important role in driving complex coacervation. Rather, our study shows that entropy driven complex coacervation can also be achieved, experimentally and computationally, without considering the effect of counterion release. Those two schools of thoughts can exist in parallel. We clarify that our study does not explicitly study the relative importance of these two factors. We believe that this answer should satisfy both the editor and reviewer #1.

"Our study does not exclude the potential role of entropy gain from counterion-release—the currently widely accepted hypothesis—as an additional driver of CC. Rather, our study shows that entropy gain from water-release is a major contributor for CC that can be further amplified in the presence of highly hydrophilic and neutral crowders such as PEG, and can be sufficient to reproduce entropy driven CC. However, our study did not examine nor directly compare the effect of counterion-release as an additional factor."